# Research on Biogas Yield from Macroalgae with Inoculants at Different Organic Loading Rates in a Three-Stage Bioreactor

**DOI:** 10.3390/ijerph20020969

**Published:** 2023-01-05

**Authors:** Alvydas Zagorskis, Regimantas Dauknys, Mantas Pranskevičius, Olha Khliestova

**Affiliations:** 1Research Institute of Environmental Protection, Vilnius Gediminas Technical University, 10223 Vilnius, Lithuania; 2Department of Environmental Protection and Water Engineering, Vilnius Gediminas Technical University, 10223 Vilnius, Lithuania; 3Department of Primary Science Institute of Modern Technologies, Pryazovskyi State Technical University, 87555 Mariupol, Ukraine

**Keywords:** anaerobic digestion, macroalgae, biogas yield, three-stage bioreactor, organic load rate

## Abstract

Macroalgae can be a viable alternative to replace fossil fuels that have a negative impact on the environment. By mixing macroalgae with other substrates, higher quality biogas can be obtained. Such biogas is considered one of the most promising solutions for reducing climate change. In the work, new studies were conducted, during which biogas yield was investigated in a three-stage bioreactor (TSB) during the anaerobic digestion of *Cladophora glomerata* macroalgae with inoculants from cattle manure and sewage sludge at different organic loading rates (OLR). By choosing the optimal OLR in this way, the goal was to increase the energy potential of biomass. The research was performed at OLRs of 2.87, 4.06, and 8.13 Kg VS/m^3^ d. After conducting research, the highest biogas yield was determined when OLR was 2.87 Kg VS/m^3^ d. With this OLR, the average biogas yield was 439.0 ± 4.0 L/Kg VS_added_, and the methane yield was 306.5 ± 9.2 L CH_4_/Kg VS_added_. After increasing the OLR to 4.06 and 8.13 Kg VS/m^3^ d, the yield of biogas and methane decreased by 1.55 times. The higher yield was due to better decomposition of elements C, N, H, and S during the fermentation process when OLR was 2.87 Kg VS/m^3^ d. At different OLRs, the methane concentration remained high and varied from 68% to 80%. The highest biomass energy potential with a value of 3.05 kWh/Kg VS_added_ was determined when the OLR was 2.87 Kg VS/m^3^ d. This biomass energy potential was determined by the high yield of biogas and methane in TSB.

## 1. Introduction

Biofuel production that utilizes renewable natural resources is one of the most promising areas of renewable energy. In order to increase biofuel production, it is necessary to develop technologies capable of generating large amounts of renewable natural resources in a short period of time [1,2]. Biofuel production from biomass ranks fourth in the world. This proves that the use of biomass in the application of various technologies is becoming more and more relevant. The desire to use macroalgae for biofuel production encourages the reduction in emissions of carbon dioxide (CO_2_), nitrogen oxides (NO_x_), sulfur dioxide (SO_2_), and particulate matter (PM) into the ambient air [3]. According to literature analysis, algae is considered one of the most promising raw materials for the production of bioenergy. Algae can be used to produce biogas, biomethane, and hydrogen [4,5,6]. While there were studies using marine macroalgae, the use of freshwater macroalgae for biogas production is not extensively studied.

The organic loading rate (OLR) is one of the main bioreactor parameters that determines the biogas yield. The use of macroalgae—growing in bodies of water and rich in organic matter—for biogas production is promising. Algae are an excellent alternative to replace high-energy crops such as Miscanthus silage used in anaerobic digestion. A biogas yield of 628.0 mL/g VS was achieved by mixing algae with Miscanthus silage at an OLR of 2.0 g VS/L d. OLR is one of the factors determining the level of substrate fermentation stabilization [7]. As the climate warms, water blooms become an increasingly pressing issue. As a result of intensive human activity in both the industrial and agricultural sectors, increasing amounts of nitrogen and phosphorus compounds enter water bodies. Organic materials promote aquatic blooms. The primary chemical elements that affect the growth of algae are carbon, nitrogen, and phosphorus [8].

Scientific research proved that high biogas yields can be obtained using anaerobically digesting algae. Studies showed that mixing cattle and chicken manure with *Chlorella pyrenoidosa* algae in a 2:1:2 ratio in a batch bioreactor increases the methane yield to 68% [9]. In a continuous bioreactor, the methane yield could be increased after optimizing the OLR.

Aquatic biomass has an advantage over other plants, as it does not compete with food crops for arable land areas. Algae produce a large increase in biomass, and after they are collected from water bodies, the condition and quality of those water bodies is improved. Biogas production does not require any additional processing of macroalgae. Depending on the environmental conditions and the type of macroalgae, macroalgae are characterized by high concentrations of carbohydrates (up to 60%), proteins (10–47%) and fats (1–3%), so they can be used as an energy source for microorganisms [10]. The highest concentration of hydrogen required for energy extraction was achieved by anaerobic digestion of algae from carbohydrates and fats [6]. Additionally, when algal biomass is mixed with other substrates, the methanogenesis process remains stable. Studies showed that by mixing *Chlorella vulgaris* with wheat silage, a methane yield of 330 mL CH_4_/g TS can be obtained. Methane concentration was 57–67% when OLR was 0.5 g VS/L d [11]. By choosing the optimal OLR, it is possible to achieve a higher methane concentration and, at the same time, a higher methane yield. For these reasons, it is recommended in biogas production to mix macroalgae with other nutrient-rich organic waste [12]. However, inoculation of aquatic plants into a substrate suitable for anaerobic digestion does not always increase methane yield. For example, when mixing *Elodea canadensis* with dairy manure, the methane yield decreased from 192.3 mL CH_4_/g VS to 151.1 mL/g VS [13].

High biogas yields can be obtained by mixing algae with cattle manure and other substrates. When the initial OLR was 5.0 Kg VS/m^3^, there was high biogas yield with a maximum value of 1058.8 L/Kg VS. In the study, the algae *Platymonas subcordiformis* was mixed with cattle manure and corn silage. However, a higher concentration of methane was obtained by introducing *Arthrospira plantensis* algae [14]. Sea grasses can contribute to higher methane yields and, at the same time, higher energy potential, which can increase up to 34% [15]. Many authors used only the original OLR in their work.

C:N ratios must be taken into account when choosing OLR. In anaerobic bioreactors, it should reach between 20:1 and 30:1. A higher ratio is not recommended, as macroalgae are characterized by a higher protein content, which can promote the growth of free ammonia and volatile fatty acids. This can have a negative effect on the development of methanogenic bacteria [16]. Other scientists ensure the required pH and C:N ratio of the substrate by inserting buffer solutions, such as CaCO_3_, which support a C:N ratio of 25:1 and improve methane yield by up to 24% [17]. Biogas yield increases when inoculants containing methanogenic bacteria are added to the substrate. Research showed that mixing macroalgae with inoculants such as sewage sludge and cattle manure can achieve high biogas yields [18]. Scientists showed that cattle manure can be used as an excellent substrate for anaerobic treatment as well. It contains many nutrients that are suitable for the growth of anaerobic microorganisms [19].

The energy value of biogas (methane content and concentration) directly depends on the amount of organic matter and microbiological processes [20]. Microorganisms of two groups (acidic and methanogenic bacteria) participate in the process of methane fermentation in the following stages: hydrolysis, acidogenesis and acetogenesis, and methanogenesis [21]. During these stages, organic matter is converted into methane (CH_4_), carbon dioxide (CO_2_), and digestate [10]. Many different microorganisms are involved in these processes, but the main group is methanogenic bacteria *Methanobacterium* sp. [22]. Studies showed that *Methanosarcina*, *Methanobacterium*, and *Methanosaeta* bacteria were highly active when mixing algae with food waste in a two-stage anaerobic bioreactor. They successfully converted acetate and hydrogen into methane [23]. Too high OLR reduces methane yield, so it can be assumed that the activity of these bacteria depends on OLR. There is no single optimal OLR value for anaerobic digestion of all macroalgae. Different authors obtain conflicting results. Using *Macrocystis pyrifer* macroalgae cultures, the highest biogas yield was obtained when the OLR was 3.00–8.00 Kg VS/m^3^ d, when it exceeded 12.00 kg VS/m^3^ d, the biogas yield decreased [24]. Other authors obtain a high yield of biogas when OLR reaches up to 2.5 Kg VS/m^3^ d [25].

OLR depends on the content of total solids (TS) and volatile solids (VS) in the substrate. The optimal concentration of TS in the substrate should not exceed 10% [26,27]. Continuous multi-stage bioreactors are not sufficiently studied. Bioreactors of periodic operation of a small capacity, the volume of which can reach from 500 mL to 2500 mL, are usually used for conducting research [27,28,29].

The life cycle assessment showed that mixing algae with other substrates in multi-stage bioreactors is energetically promising. The ratio of system energy consumption to energy production reached 0.24 when the substrate was processed from a mixture of algae and food waste in a two-stage bioreactor [5].

OLR is an important parameter that can increase biogas and methane yields, but too high OLR can reduce them. This paper presents new research aimed at investigating the biogas yield in a three-stage bioreactor by anaerobic digestion of *Cladophora glomerata* macroalgae with inoculants such as cattle manure and sewage sludge under different OLRs to increase the energy potential of the biomass.

## 2. Materials and Methods

### 2.1. Substrate and Inoculum

Green macroalgae *Cladophora glomerata* (MA) was mixed with cattle manure (CM) and sewage sludge (WS) for the research. CM and WS were used as a source of activated methanogenic bacteria. Macroalgae were taken from a freshwater surface water body—a Šventoji river. A *Cladophora glomerata* macroalgae culture was selected for the research due to its large population. The green macroalgae *Cladophora glomerata* is widely distributed in the Baltic, Black, Azov Seas, and other regions, and forms floating mats in the water. It grows abundantly in both sea and fresh water [30,31,32]. This species was used in this type of research for the first time to determine the biogas yield from different OLR.

Cattle manure was taken from a cattle farm, while activated anaerobic sludge was taken from a city sewage treatment plant. Before conducting the tests, the moisture content (DS), total solids (TS), volatile solids (VS), and inorganic solids (NS) of each raw material were determined. Tests for these parameters were performed using standardized methods [33]. In order to measure TS, a 5 g sample was taken and dried at 105 °C for 12 h. DS was determined knowing that it was equivalent to the humidity reading. For VS determination, the sample was placed in a muffle furnace and heated at a temperature of 550 °C for 1 h. The mass lost is equal to VS and the mass remaining is equal to NS [26,34]. The physical parameters of MA and inoculum are presented in Table 1.

In order to select the optimal mixture of MA, CM, and WS, the elemental composition of each biomass was determined. The elemental composition of biomass and substrate was determined using the elemental composition analyzer EA 3000. The obtained research results are presented in Table 2.

According to literature, the recommended C:N ratio in the substrate should be between 1:20 and 1:30. During the research, the biomass was mixed in such a ratio that the C:N ratio reached 26. At this ratio, the optimal biogas yield and methane concentration are achieved.

The wet mass (DM) of MA and CM was ground with a grinder. It was then weighed with nets and, after determining the physicochemical parameters, was mixed with the WS crop. Considering the working volume of the bioreactor, a total of 300 L of substrate was prepared. Biomass mixing ratios and physicochemical parameters are presented in Table 3.

The substrate presented in Table 3 was mixed and homogenized in the preparation chamber of the three-stage bioreactor. After mixing everything, the physicochemical parameters of the substrate were additionally determined. The values of the physicochemical parameters of the substrate are presented in Table 4.

### 2.2. Description of Laboratory Stand

A three-stage bioreactor (TSB) pilot prototype was used for the research. The device consists of three chambers: a primary preparation chamber (capacity: 60 L), a secondary anaerobic chamber (capacity: 60 L), and a tertiary anaerobic chamber (capacity: 300 L). The TSB stand is shown in Figure 1.

In PC-I, the biomass is crushed, mixed, and homogenized with a coarse fraction organic waste shredder. From the preparation chamber (PC-I) biomass is fed to AC-II. A transitional process takes place in the AC-II chamber when the substrate transitions from an aerobic (saturated with dissolved oxygen) state to an anaerobic state. In the chamber, the biomass is mixed and heated to a temperature of 20 °C. From AC-II, the substrate is fed to AC-III, where mesophilic conditions are maintained. Studies showed that methanogenesis works efficiently when the AC-III temperature is maintained at 37 °C [32,33,34]. The substrate from AC-III to AC-III does not adversely affect the methanogenic bacterial colonies that exist in AC-III.

The substrate is then mixed using a mechanical mixer and heated at 300 W/200 V. An industrial thermostat is used to adjust to the desired temperature. For this purpose, thermocouples were inserted into AC-II and AC-III in order to have contact with the substrate in the camera [34]. The mixer is equipped to mix AC-II and AC-III substrate at the same time. From AC-III, the substrate is removed through an outlet. Biogas from AC-II and AC-III is fed into the biogas storage tank for elemental composition analysis.

### 2.3. Experimental Setup and Procedure

The substrate is mixed and homogenized in the bioreactor PC-I. From PC-I, heating takes place in the AC-II chamber. From AC-II, the substrate is supplied via autophagy to AC-III. Since the volume of PC-I and AC-II is 60 L, the filling of AC-III with a volume of 300 L was completed in five increments of 60 L each. The research was conducted in three stages. During all phases, temperatures of 20.0 °C (in AC-II) and 37.0 °C (in AC-III) were maintained in the TSB chambers of continuous operation. A mixer installed in the TSB simultaneously mixed the substrate in AC-II and AC-III around the clock at a constant rotation speed of 5.0 rpm. The substrate supply modes were calculated so that 300 L of substrate flow through the AC during one mode. The retention time (RT) and organic load rate (OLR) parameters important for biogas and methane yield were determined taking into account the VS concentration of AC-III supplied substrate and the selected supply capacity (SC). The total volume of substrate (TVS) was 300 L. RT and OLR were calculated using Equations (1) and (2), respectively. The characteristics of research stages are presented in Table 5.

This amount of substrate is equal to the working volume of AC-III. After loading, the AC-III substrate supply was not carried out until the biogas yield stabilized. After the biogas yield stabilized, the tests were carried out in four stages. During the first stage, the OLR was equal to 0. During the second, third, and fourth stages, the OLR reached 2.87, 8.13, and 4.06 Kg VS/m^3^ d, respectively. OLR was ensured when the supply capacity (SC) reached 21.4, 60.0, and 30 L/d, respectively. When the biogas yield stabilized after 14 days, the OLR of the supplied substrate reached 2.87 Kg VS/m^3^ d. At this OLR, the estimated residence time (RT) of the substrate is 14 days. During phase II, after the biogas yield stabilized, the SC of the substrate was increased to 60 L/d, and the OLR to 8.13 Kg VS/m^3^ d (phase III). After the stabilization of the biogas yield during phase III, SC was reduced to 30 L/d, and the OLR to 4.06 Kg VS/m^3^ d. This supply mode (phase IV) was ensured until the biogas yield stabilized. Substrate supply and removal took place every day at the same time of day.

TS, VS, NS, C, N, H, S, and pH parameters were determined in the substrate before and after each stage. TS, VS, and NS were determined using standardized methods [33]. C, N, H, and S were determined using an elemental composition analyzer EA 3000, and pH parameters were measured using a Mettler Toledo MultiSeven meter. Biogas yield and qualitative composition were determined every day. When evaluating the qualitative composition of biogas, the concentrations of CH_4_, CO_2_, O_2_, and H_2_S in the biogas were determined using the GasData series biogas composition analyzer GFM 410.

### 2.4. Analytical Methods and Statistical Analysis

TS and VS are determined based on EPA research methodology [33].

The concentrations of elements C, N, H, and S in biomass were determined using the elemental composition analyzer EA 3000. This measurement error is ±0.3%. Manufacturer: Eurovector, Pavia, Italy.

The concentration of hydrogen ions in pH was determined using a Mettler Toledo MultiSeven meter. Device error is ±0.002. Manufacturer: Mettler Toledo Solutions, Greifensee, Switzerland.

Retention time was calculated according to the following formula:(1)$$\mathrm{RT}=\frac{\mathrm{TVS}}{\mathrm{SC}}$$
where RT is retention time (d), TVS is total volume of substrate (L), and SC is supply capacity (L/d).

The organic load value (OLR) was calculated using the equation [35]:(2)$$\mathrm{OLR}=\frac{\mathrm{SC}\times \mathrm{VS}}{\mathrm{TVS}}$$
where OLR is organic loading rate (g VS/L d) and VS was the volatile solid concentration of the inflowing substrate (g VS/L).

The amount of biogas produced per day was calculated according to the following formula [36]:(3)$$V_{0}^{\mathrm{dr}}=V\times \left(\frac{\left(P-P_{W}\right)\times T_{0}}{\left(p_{0}\times T\right)}\right)$$

$V_{0}^{\mathrm{dr}}$ is the volume of dry biogas under normal conditions (L), V is the volume of biogas measured in the biogas storage tank (mL), P is the biogas pressure measured in the biogas storage tank (hPa), P_W_ is the water vapor pressure dependent on the ambient temperature (hPa), T_0_ is the normal temperature (T_0_ = 273 K), p_0_ is the normal pressure (p_0_ = 1013 hPa), and T is the temperature of the biogas released during fermentation (K).

The qualitative composition of biogas was determined according to the electrochemical research method. The concentrations of CH_4_, CO_2_, O_2_, and H_2_S were determined using the GasData series biogas composition analyzer GFM 410. The measurement limits and error of the device: 0–100% and ±3.0% (for CH_4_), 0–100% and ±3.0% (for CO_2_), 0–25% and ±0.5% (for O_2_), and 0–1500 ppm and ±5.0% (for H_2_S). Manufacturer: Gas Data Limited, Coventry, United Kingdom. Before the measurement, the device was calibrated. Periodically, the device data were compared with the data obtained during the chromatographic analysis.

The measurements were performed in three replicates. After receiving the research results of a certain parameter, its estimate was calculated, which was the arithmetic mean of the individual measurements. Knowing the arithmetic mean of the parameter, the experimental variance corresponding to the probability distribution of the individual measurements was calculated. The calculation found the best estimate of the dispersion of the arithmetic mean, which is the experimental arithmetic variance of the mean, equal to a normal distribution. After obtaining the estimate of the dispersion of the arithmetic mean, the experimental standard deviation of the arithmetic mean was calculated. The honest significant difference test was used to examine the significance of the differences between the analyzed variables. Differences were considered significant when *p* = 0.05.

The Microsoft Office Excel 2019 program (manufacturer: Microsoft, Washington, DC, USA) was used for statistical calculations. Arithmetic averages of measurements, experimental arithmetic variances, and standard deviations of the arithmetic mean were calculated using the statistical functions of Microsoft Office Excel 2019: AVERAGE, VARP and STDEV.

## 3. Results

Research prescribed biogas yields during different stages. The results of the research are presented in Figure 2. The results of the research show the active rug of the biogas output for the first four days. Such a sudden increase was potentially caused by a crop introduced into the substrate. Already on the first day of the research, a biogas yield of 496.3 L/m^3^ d was found. Four days later, the biogas yield reached its highest peak at 2585.6 L/m^3^ d. After that, it began to decline. After 14 days, the biogas yield fell below 1000 L/m^3^ d. It was not supplied to the TSB substrate (OLR = 0). During those 14 days, 500 L/Kg VS was produced.

In order to increase the yield of biogas, 1.6 Kg SM/d substrate was supplied to the TSB, which amounted to 0.87 Kg VS/d, and OLR was 2.87 Kg VS/m^3^ d. After increasing OLR, the yield of biogas during stage II increased to 1252.0 L/m^3^ d. It could be assumed that the increase was due to the fact that the methanogenic bacteria again received organic material. After the biogas yield stabilized on the 21st day, the OLR of the supplied substrate was increased to 8.13 Kg VS/m^3^ d. TS levels rose to 4.5 Kg SM/d. After increasing the amount of supplied substrate, the yield of biogas started to increase. The highest increase was found on the 22nd day. With this amount of supplied substrate, the amount of biogas produced increased to 1742.0 L/m^3^ d. After the stabilization of the biogas, the OLR of the supplied substrate was reduced to 4.06 Kg VS/m^3^ d. The TS of the supplied substrate reached 2.24 Kg SM/d, and VS reached 1.09 Kg VS/Kg. By reducing the amount of substrate, the produced biogas decreased for up to 47 research days. The amount of biogas produced was 1501.0 L/m^3^ per day. From the research results presented in Figure 2, it can be seen that when OLR was increased from 2.87 to 8.13 Kg VS/m^3^ d, the biogas yield decreased by 1.55 times.

The yield of biogas depends on the VS present in the substrate. Figure 3 shows the dependence of biogas yield on the amount of supplied and decomposed substrate VS during different stages. Each stage represents the residence time of the substrate. From the research results presented in the Figure 3, it can be seen that the highest biogas output is obtained when the substrate remains for 14 days. When the substrate is maintained for 14 days, up to 626.8 L/Kg of VS_destroyed_ biogas is extracted per day. After reducing the substrate residence time to 5 days, the biogas yield was 341.9 L/Kg VS_destroyed_ biogas per day. Research showed that by shortening the duration of the substrate, the yield of biogas from 1 Kg VS substrate decreases. Methanogenic bacteria do not manage to completely break down the substrate VS. It can be assumed that this is due to the structure of macroalgae cells, which bacteria are unable to completely disassemble. Additionally, the decrease in bacterial activity can be caused by the decreasing pH of the substrate during anaerobic digestion. Increasing the substrate residence time to 10 days resulted in an increase in the amount of degraded substrate. Biogas yield increased to 491.7 L/Kg VS_destroyed_.

After reducing the OLR of the substrate supplied per day to 4.06 Kg VS/m^3^ d, the RT of the substrate decreased to 10 days. In this case, the biogas yield during stage IV decreased from 544.9 L/Kg VS_destroyed_ to 491.7 L/Kg VS_destroyed_. It is necessary to note that during the research, the concentration of the volatile part supplied to the TSB in the substrate was between 49% and 55% during all stages. The research results show that the biogas yield depended on the RT of the substrate. The higher the RT, the higher the biogas yield.

Levels of methane yield and concentration in TSB at different OLRs are presented in Figure 4. After loading the bioreactors, the maximum methane concentration was determined after four days. On the fourth day of research, the methane concentration reached 80.4%. The methane concentration then began to decrease. By the 14th day, the methane concentration decreased to 64.2%. By feeding the substrate to the TSB, it is possible not only to increase the yield of methane, but also the concentration of methane in biogas. During stage II, when the OLR reached 2.87 Kg VS/m^3^ d (substrate TS was 1.60 Kg SM/d), the substrate methane concentration increased to 73.7%. After 20 days of the study, the methane concentration stabilized and reached an average of 73.1%. After increasing the OLR to 8.13 Kg VS/m^3^ d, both methane output and methane concentration increased to 80.8%. This methane concentration stabilized until the end of stage III. After reducing the OLR to 4.06 Kg VS/m^3^ d, the methane concentration started to decrease. After the 40th day of the study, the methane concentration stabilized and reached 74.0%. It should be noted that increasing the OLR of the supplied substrate increased the yield of methane. During stage II, it reached 903.0 L CH_4_/m^3^ d. The maximum yield of methane was reached during stage III at 1401.0 L CH_4_/m^3^ d. This was due to higher methane concentration and higher biogas yield. After reducing the OLR of the supplied substrate to 4.06 Kg SM/d, the methane concentration remained sufficiently high and reached 70%. Lower biogas yield also resulted in lower methane yield. At the end of stage IV, the yield of methane reached 1100.0 L CH_4_/m^3^ d.

As Figure 4 demonstrates, carbon dioxide concentration also depends on the amount of the substrate. In stages II and III, the CO_2_ concentration of the supplied substrate increased. By reducing the amount of the supplied substrate to 2.24 kg SM/d of CO_2_, the concentration of CO_2_ was reduced to 25.8% during stage IV. Such a change is determined by the activity of the methanogenic bacteria in the substrate.

During the anaerobic processing of the substrate, methane is released during the methanogenesis process. Carbon dioxide concentrations are released during the processes of acidogenesis, acetogenesis, and methanogenesis. As a result, increasing the content of the TSB supplied substrate increases not only in methane but also CO_2_. Studies showed that CO_2_ concentrations also change during different stages. During all stages, the CO_2_ concentration ranged from 20% to 35%. The greatest concentration of CO_2_ was determined during stage II when the duration of the served substrate was the longest at 14 days. After increasing the amount of the TSB supplied substrate to 4.50 kg SM/d, the CO_2_ concentration in the biogas decreased to 20%.

Studies found that oxygen (O_2_) and hydrogen sulfide (H_2_S) concentrations were low and at all stages of the experiment reached 0–2% and 0–10 ppm, respectively.

The yield of biogas depends on the duration of the substrate. The biogas yields are delayed based on the duration of the substrate in Figure 5. The duration of the substrate depends on the amount of substrate supplied to the TSB. During the research, a high correlation was found between the biogas yield and the duration of the substrate’s presence. The correlation coefficient determined during the research reached 0.90.

As OLR increases, the residence time of the substrate in the bioreactor decreases. It can be assumed that methanogenic bacteria are unable to break down VS. Short processes do not allow methanogenesis to occur and affect the biogas yield. This is shown by the obtained research results in Figure 5.

When OLR was 8.13 Kg VS/m^3^ d (RT = 5 days), the average biogas output reached 306.3 L/Kg VS_destroyed_. After reducing the OLR to 4.06 Kg VS/m^3^ d, the retention time of the substrate increased to 10 days. Biogas output per day increased to 480.0 L/Kg VS_destroyed_. The highest biogas yield was determined when the RT of the substrate reached 14 days (OLR = 2.87 Kg VS/m^3^ d). The TSB biogas yield was 624.6 L/Kg VS_destroyed_.

In Table 6, the average and maximum yields of biogas and methane are presented. The data in the table are presented in order to evaluate the yield dependence on VS_added_.

The highest biogas and methane yield was in Stage II, when the OLR was 2.87 Kg VS/m^3^ d. The average yield of biogas per day was 439.0 ± 4.0 L/Kg VS_added_, and the methane yield was 306.5 ± 9.2 L/Kg VS_added_. After increasing OLR, biogas and methane yields decreased. In fact, the lowest yields were determined when OLR was 8.13 kg VS/m^3^ d. Biogas and methane yields were 198.7 ± 2.2 L/Kg VS_added_ and 154.1 ± 5.4 L/Kg VS_added_, respectively.

Anaerobic digestion (AD), a multistep process consisting of hydrolysis, acidogenesis, acetogenesis, and methanogenesis steps, was carried out by specialized microorganisms. Archaea, *Methanomicrobiales*, *Methanomassiliicoccales*, and *Methanosarcinales* bacteria dominate the methane production process [37]. It is a homogeneous process and these microorganisms are involved in all stages. The activity of microorganisms is inhibited by external factors, such as volatile fatty acids, pH, salinity, and phenols. Increasing OLR increases the concentration of volatile fatty acids. Therefore, we can assume that volatile fatty acids had the greatest influence on the reduction in methane yield, which, when the concentration exceeded 20.00 g/L, inhibited methanogenic bacteria [38]. The pH of the substrate was higher than 6.5, and the macroalgae were taken from fresh water of the river, so the influence of salinity and phenols was minimal.

The research revealed the dependence of the substrate TS and VS on the substrate maintenance duration. The substrate TS and VS dependence on the substrate maintenance duration is shown in Figure 6. The data provided shows that the decrease in the substrate TS is linked to the duration of the substrate in AC-III. The maximum decrease in the TS and VS occurred when the OLR was 2.84 kg VS/m^3^ d (SC reached 21.4 L/d). With the substrate maintenance duration of 14 days, TS and VS decreased by 35.3%. Reducing the duration to 5 days resulted in a TS reduction and VS destroyed of 12.6%. Studies showed that increasing the duration of the substrate achieves a higher efficiency of TS and VS. After serving a large amount of substrate (SC = 60 l/d), the methanogenic bacteria cannot break down the substrate TS and VS, so the percentage of breakdown of these parameters is lower.

The research results show that a high breakdown of TS and VS is achieved when the substrate maintenance duration is 10 days. This is when supply capacity (SC) reaches 30.0 L/d (OLR = 4.06 kg VS/m^3^ d), which is 10% of the total bioreactor loading mass. With such a SC substrate, the breakdown was 31.1%.

In order to evaluate changes in the elementary composition of the substrate at different OLRs, research was conducted by measuring concentrations of C, N, H, and C supplied to different OLR fermented substrates. Substrate elemental composition studies are given in Table 7.

One of the main parameters of the elemental composition is the C:N ratio. When using a continuous bioreactor, it is necessary to maintain an optimal C:N ratio. The C concentration of the substrate supplied to TSB reached 650 ± 19.5 g/Kg TS, the N concentration reached 25.0 ± 0.1 g/Kg TS. In the study, the feed substrate was maintained at a C:N ratio of 26:1. As a result of this research, it was found that at different OLRs, the concentrations of C, H, and S decreased, while the concentration of N increased. The greatest elemental change is found at the highest OLR, while the smallest change occurred during stage I when there was no substrate supplied to the TSB. When the OLR reached 2.87 Kg VS/m^3^ d, ΔC reached 15.1 ± 0.03 g/Kg TS d. After increasing the OLR to 8.13 g/Kg TS d, ΔC decreased to 12.5 ± 0.04 g/Kg TS d. ΔH decreased from 0.7 ± 0.003 g/Kg TS d to 0.5 ± 0.007 g/Kg TS d, respectively. The concentration of ΔS also fluctuated between 0.3 ± 0.001 g/Kg TS d and 0.2 ± 0.001 g/Kg TS d depending on the OLR. At a higher OLR, a decrease in C, H, and S occurred, which resulted in a lower biogas yield from Kg VS_added_. During the process of methanogenesis, during the decomposition of organic materials, the elements C, H, and S present in the substrate pass from the liquid to the gaseous phase. The highest nitrogen levels were obtained when the OLR was 2.87 Kg VS/m^3^ d. As the OLR increased, the concentration of nitrogen decreased. As a result of this research, it was found that when the OLR was increased from 2.87 Kg VS/m^3^ d to 8.13 Kg VS/m^3^ d, ΔN decreased from 0.4 ± 0.003 to 0.2 ± 0.003 g/Kg TS d. From the obtained research results, it can be seen that the nutrient properties of the substrate and the suitability of the fermented substrate for soil fertilization depend on the OLR. The pH value of the substrate during different stages was close to neutral and reached 7.07–7.77.

The energy potential was determined after the research. The highest yields of biogas and methane were determined when OLR was 2.87 Kg VS/m^3^ d. The average methane concentration was 69.7%. The energy value of methane is 35.8 MJ/m^3^ [39]. In this case, a high energy value of biogas was achieved, which was 25.0 MJ/m^3^. When the methane concentration is 97%, 9.67 kWh of energy can be extracted from 1 m^3^ of biogas [40]. In this case, the determined amount of energy was 6.95 kWh/m^3^, 0.439 m^3^ of biogas was produced from 1 Kg VS_added_, and the energy potential of the substrate reached 3.05 kWh/Kg VS_added_. The energetic value of biogas when OLR was 8.13 and 4.06 Kg VS/m^3^ d energy potential reached 1.54 and 2.81 kWh/Kg VS_added_, respectively. The average energy consumption of the three-stage bioreactor for mixing and heating was 0.17 and 1.80 kWh/d, respectively. Total energy consumption was 1.97 kWh/d or 0.69 kWh/Kg VS d. The ratio of system energy consumption and produced energy reached 0.23.

## 4. Discussion

During the conducted research, it was found that the OLR has a significant influence on biogas yield. The highest biogas output occurred when the OLR reached 2.87 Kg VS/m^3^ d. The average yield of biogas and methane was 439.0 ± 4.0 L/Kg VS_added_ and 306.5 ± 9.2 L CH_4_/Kg VS_added_, respectively. The lowest average yield of biogas and methane was 198.7 ± 2.2 L/Kg VS_added_ and 154.1 ± 5.4 CH_4_/Kg VS_added_ (OLR = 8.13 Kg VS/m^3^ d).

The research conducted in this study showed that the best biogas yield was obtained when the OLR was 2.87 Kg VS/m^3^ d. By mixing macroalgae with fruit waste, a high biogas yield was obtained when the OLR was 3 Kg VS/m^3^ d. Biogas yields using macroalgae cultures vary widely in scientific literature. Depending on the OLR, the biogas yield can reach anywhere from 46 mL/g VS_added_ to 1200 mL/g VS_added_. Methane yield can reach anywhere from 28 mL/g VS_added_ to 900 mL CH_4_/g VS_added_. Studies were conducted on biogas and methane yields when macroalgae were mixed with other substrates, such as fruit waste and chicken manure. Some studies showed that when mixing macroalgae with fruit waste at a ratio of 50:50, the yield of methane was 46 mL CH_4_/g VS. During the research, OLR was 3 Kg VS/m^3^ d. By increasing the amount of fruit waste to 70%, a lower methane yield was achieved. It reached 28 mL CH_4_/g VS. The inoculant was chicken manure [41]. Meanwhile, in this study, a methane yield of 306.5 ± 9.2 L CH_4_/Kg VS_added_ was achieved by mixing green macroalgae with cattle manure and sewage sludge. The yield of methane was even six times higher. This shows that using cattle manure and sewage sludge as inoculant results in higher methane yields. Research by other scientists showed that the maximum biogas yield was achieved when the mixing ratio of macroalgae and fruit waste reached 60:40. The maximum biogas yield achieved was 295 mL/g VS, and the methane concentration was 72%. The study was conducted for 45 days under mesophilic conditions [18]. During this study, a higher biogas yield was achieved, which reached 439.0 ± 4.0 L/Kg VS_added_. Green macroalgae performed better compared to marine macroalgae of different species. A methane yield of 396 mL/g VS was achieved by mixing algae with cattle manure. The inoculant was anaerobic sewage sludge, and OLR was 1.3 g VS/L d [42].

Compared to this study, high biogas yields were found using *Ulva rigida* macroalgae cultures. This research found that the highest biogas yield was achieved when the OLR was 1.66 g VS/L d. When macroalgae was mixed with sugar waste, maximum biogas yield was 1200 mL/g VS_added_. The maximum biogas yield was achieved using green macroalgae at a level of 114 mL/g VS. Methane concentration was 75% [29].

In this study, the biogas yield was 2.7 times lower. By using sugar waste, the organic materials are transformed into biogas [43]. Up to 90% of biodegradable waste mass is converted to biogas. The energy value of biogas (methane content and concentration) depends on the amount of organic matter and the microbiological processes involved [44]. Scientists found that algae culture, such as *Ulva* sp., reduces the concentration of volatile fatty acids, and thus improves the quality of the biogas. In this study, the lower biogas yield was due to lower VS breakdown and conversion to biogas. A number of biogas yield studies using macrocystis were carried out by other researchers. Scientists used *M. pyrifera* for fermentation and received 282 mL/g VS to 383 mL/g VS biogas yield [45]. Compared to this study, the biogas yield was 1.2 times lower. The yield of biogas depends on the amount of VS in the substrate. Therefore, the yield of biogas can be judged from a broken-down substrate VS. Macroalgae were mixed with sugar waste and an inoculate of sewage sludge, which resulted in an 84.2% VS breakdown. The methane yield was 600 mL/g VS_added_. The Zhong bioreactor used blue macroalgae. Using these macroalgae was about 50% [46]. In this study, the decomposition of VS was lower and reached 35.3%, and the yield of methane was also lower than 600 mL/g VS_added_ and was 306.5 ± 9.2 L CH_4_/Kg VS_added_. We can assume that worse decomposition could be caused by stronger cell walls of the green macroalgae *Cladophora glomerata*, which is why the bacteria could not fully assimilate VS.

In this study, the high yield of biogas and methane could be due to the optimal C:N ratio. In this work, the feed substrate was maintained at a C:N ratio of 26:1. In this ratio, the average methane concentration was 73.1% to 80.8% at different stages. A maximum methane yield of 306.5 ± 9.2 L CH_4_/kg VS_added_ was found when the OLR was 2.87 Kg VS/m^3^ d. In this research, the supply substrate maintained a C:N ratio of 26:1. This ratio was maintained by mixing the biomass of different components to form a combination of MA, CM, and WS. Researchers reached the highest yield of methane in an anaerobic bioreactor by treating blue macroalgae at a C:N ratio of 20:1. The methane yield was 325 mL/g VS, and the methane concentration was 61.7% [46]. A lot of work was performed on a continuous operation by scientists Baltrėnas and Kolodynskij, utilizing bioreactors. They used one of the first two- and three-stage bioreactors in their studies, researching biogas output in the mixing of green waste with food and animal waste and biochar. Their studies showed the effect of bioreactor stages on biogas output [47,48].

Using a three-stage bioreactor, the highest methane yield was achieved when OLR was 2.87 Kg VS/m^3^ d. The yield of methane in the bioreactor was 306.5 ± 9.2 L CH_4_/Kg VS_added_. Other scientists sought to utilize a two-stage bioreactor for estimating methane yield. The researchers used *Napier* plants to estimate biogas yield. The OLR reached 0.5 Kg VS/m^3^ d, while the methane yield in the one-stage bioreactor reached 282 L CH_4_/g VS. In the two-stage bioreactor, methane yield was 30% higher than in the single-stage bioreactor [49]. Based on these research results, we can assume that the three-stage bioreactor had a 38% higher biogas yield than the one-stage bioreactor. The better effect could be due to the more stable methanogenesis process supported in the three-stage bioreactor.

In this study, it was proved that increasing the OLR does not always increase the biogas yield. In this study, biogas yield decreased when OLR was at both 4.06 Kg VS/m^3^ d and 8.13 Kg VS/m^3^ d. Similar results were obtained by other researchers. When using *Macrocystis pyrifer* macroalgae cultures, when the OLR exceeded 6.85 kg VS/m^3^ d, the biogas yield suddenly decreased [24]. It can be assumed that the methane yield decreased due to the decreased activity of *Methanomicrobiales*, *Methanomassiliicoccales*, and *Methanosarcinales* bacteria. It can be assumed that the yield of biogas could be inhibited by volatile fatty acids and unfavorable pH [37].

During the research presented in this paper, a high yield of biogas was obtained when the TS of the supplied substrate, consisting of macroalgae and inoculants, reached 8.1%. The total amount of inoculants consisting of CM and WS reached 2.8% and 1.3%, respectively. Studies showed that it is recommended that TS should not exceed 10%. At higher TS concentrations, biogas yield decreases, which was confirmed by other studies as well [50,51,52].

The results of the research show that the use of aquatic plants that promote water bloom for biogas production is a suitable alternative to fossil fuels because it has a high energy potential that reaches 3.05 kWh/Kg VSadded biomass. Currently, more and more attention is paid to renewable energy. The development of renewable energy production was motivated by the fact that fossil fuel technologies are polluting, expensive, can only be mined in certain places, and their resources are limited. Use for energy production is a good alternative to fossil fuels. Excess biomass of macroalgae floating in water bodies or washed ashore reduces the need for fertile areas for biogas production. In this way, larger areas can be used for growing food crops.

## 5. Conclusions

Greenhouse gas emissions can be reduced by intelligent management of biomass resources. Therefore, it is necessary to look for sources of biomass intended for energy purposes that meet economic and ecological criteria. Unwise management of biomass resources can lead to negative consequences for the environment and people, during which environmental pollution would increase. Improperly used land areas would negatively affect food supply and prices. Studies showed that the use of macroalgae for energy extraction is a viable alternative to typical energy crops.

In this study, we investigated the yield of biogas and methane during the anaerobic digestion of green macroalgae *Cladophora glomerata* with inoculants consisting of cattle manure and sewage sludge. Biogas and methane yields at different OLRs were recorded during the research. This research revealed that the highest biogas yield was obtained when the OLR was 2.87 Kg VS/m^3^ d. At this OLR, biogas yield was 439.0 ± 4.0 L/Kg VS_added_, methane yield was 306.5 ± 9.2 L CH_4_/Kg VS_added_, and methane concentration was 80%. After increasing the OLR to 8.13 Kg VS/m^3^ d, the biogas yield decreased to 198.7 ± 2.2 L/Kg VS_added_. This research also determined the yield of biogas and methane at different stages of substrate supply to TSB. This study showed that after increasing the OLR from 2.87 Kg VS/m^3^ d to 8.13 Kg VS/m^3^ d, biogas yield decreased by 1.55 times. A high concentration of methane was found at different OLRs, reaching a range of 68% to 80%. Our research shows that by controlling the OLR of the substrate, we can obtain a higher yield of biogas.

The novelty of this work is that the biogas yield was investigated in a three-stage bioreactor by anaerobic digestion of *Cladophora glomerata* macroalgae with inoculants from cattle manure and sewage sludge under different OLRs.

The anaerobic digestion process depends on the substrate OLR and significantly influences the biogas yield when green macroalgae with inoculants from cattle manure and sewage sludge are used in TSB. The use of green macroalgae for biogas production is a viable alternative to fossil fuels. The results of this study show that a mixture of *Cladophora glomerata* and inoculants at optimal OLR is a sustainable and promising method for efficient energy extraction. At the optimal OLR, not only the yield of biogas increased, but also the concentration of methane, which led to a higher yield of methane, a high energy value of biogas and the energy potential of the substrate, which reached 3.05 kWh/Kg VS_added_.

## Figures and Tables

**Figure 1 ijerph-20-00969-f001:**
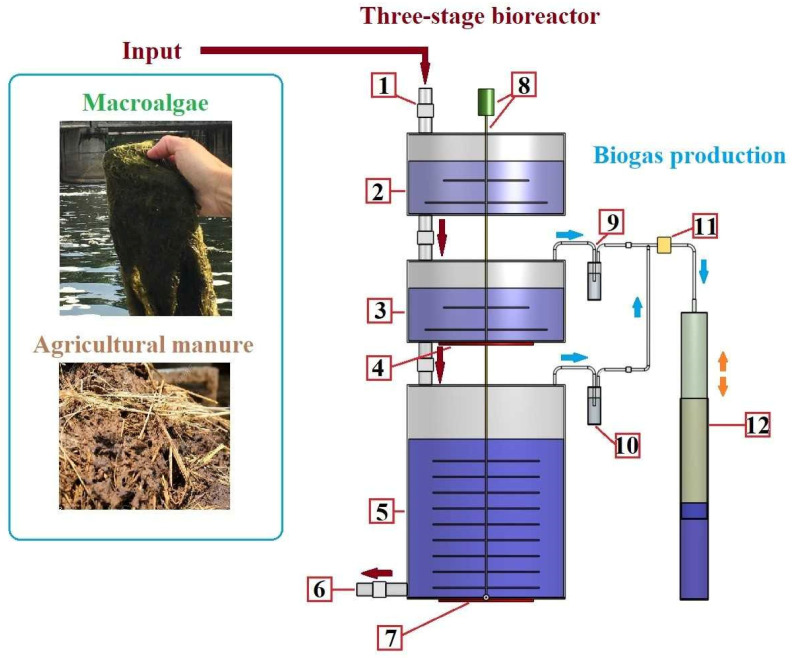
A three-stage bioreactor stand: 1—substrate supply pipe to the primary preparation chamber; 2—primary preparation chamber (PC-I); 3—secondary anaerobic chamber (AC-II); 4—AC-II heating elements; 5—tertiary anaerobic chamber (AC-III); 6—substrate outflow pipe; 7—AC-II heating elements; 8—mixer with motor; 9 AC-II biogas condenser; 10—AC-III biogas condenser; 11—shutter; and 12—biogas storage capacity.

**Figure 2 ijerph-20-00969-f002:**
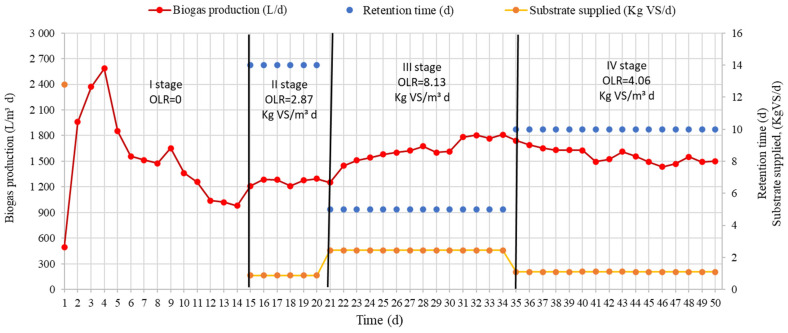
Dependence of biogas production on residence time and OLR.

**Figure 3 ijerph-20-00969-f003:**
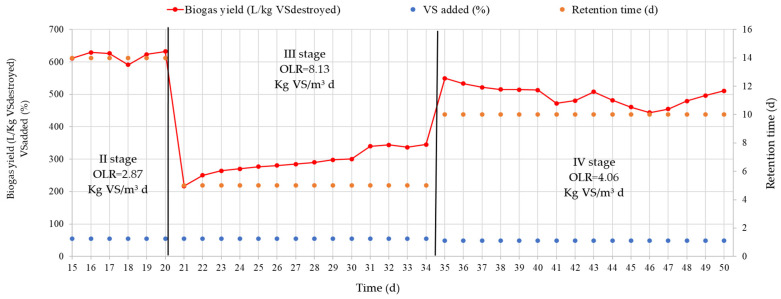
Biogas yield from the supplied amount of substrate VS_destroyed_ and retention time at different OLR.

**Figure 4 ijerph-20-00969-f004:**
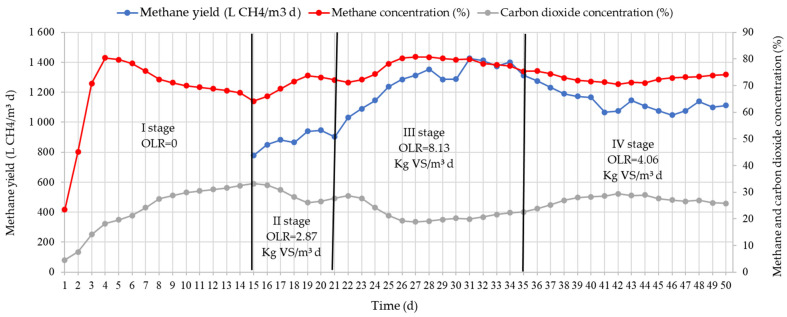
Levels of methane (CH_4_) yield and carbon dioxide (CO_2_) concentration in TSB at different OLRs.

**Figure 5 ijerph-20-00969-f005:**
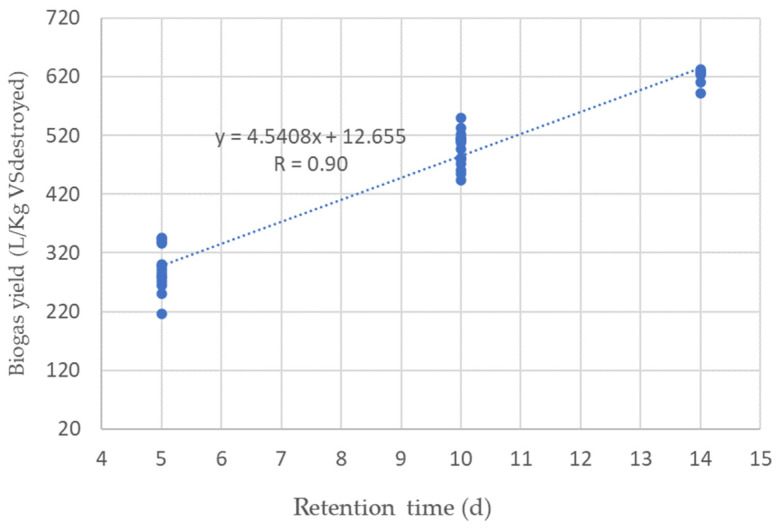
Dependence of biogas yield on the substrate retention time (RT) in TSB.

**Figure 6 ijerph-20-00969-f006:**
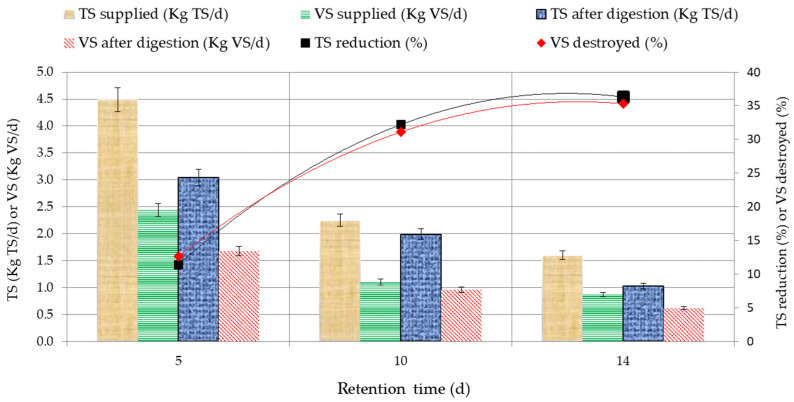
Dependence of the substrate TS and VS on retention time.

**Table 1 ijerph-20-00969-t001:** Physical parameters of MA and inoculum.

Biomass	DS, %	TS, g/Kg	VS, g/Kg	VS, %	NS, g/Kg
MA	66.7 ± 0.4	333.2 ± 0.4	125.4 ± 0.3	37.6 ± 0.3	207.08 ± 0.5
CM	80.8 ± 0.2	192.0 ± 0.2	132.5 ± 0.4	69,0 ± 0.2	59.5 ± 0.3
WS	93.1 ± 0.3	69.3 ± 0.3	35.5 ± 0.3	51.2 ± 0.4	33.9 ± 0.4

**Table 2 ijerph-20-00969-t002:** Chemical parameters of MA and inoculum.

Biomass	C, %	N, %	H, %	S, %
MA	86.6 ± 0.2	0.53 ± 0.01	4.70 ± 0.01	0.8 ± 0.1
CM	62.6 ± 0.2	4.44 ± 0.01	5.30 ± 0.02	3.4 ± 0.2
WS	48.0 ± 0.1	3.0 ± 0.02	–	–

**Table 3 ijerph-20-00969-t003:** The composition of the substrate supplied to the TSB.

Biomass	VS, Kg	TS, Kg	DM, Kg	Density, Kg/L	DM, L
MA	4.10	10.9	71.3	1.01	70.6
CM	5.37	7.8	50.7	0.91	55.7
WS	1.90	3.7	55.0	1.02	53.9
Water	–	–	–	–	119.8
Total	11.37	22.38	296.8	–	300.0

**Table 4 ijerph-20-00969-t004:** Physicochemical parameters of the substrate.

	TS, g/L	VS, g/L	NS, g/L	C/N	pH
Substrate	80.91 ± 0.32	40.42 ± 0.33	40.51 ± 0.31	26	7.77

**Table 5 ijerph-20-00969-t005:** Three-stage bioreactor performance parameters.

Stage	Period of Operation, d	VS, g/L	TS_added_, Kg SM/d	VS_added_, Kg VS/d	TVS, L	SC, L/d	RT, d	OLR, Kg VS/m^3^ d
I	1–14	40.40	–	–	–	–	–	–
II	15–20	40.23	1.60	0.87	300.0	21.4	14	2.87
III	21–34	40.65	4.50	2.44	300.0	60.0	5	8.13
IV	35–50	40.60	2.24	1.09	300.0	30.0	10	4.06

**Table 6 ijerph-20-00969-t006:** Average and maximum yield of biogas and methane.

Stages	OLR, Kg VS/m^3^ d	Biogas Yield, L/Kg VS_added_		Methane Yield, L CH_4_/Kg VS_added_	
		Avg.	Max.	Avg.	Max.
I	–	500 ± 5.0 ^1^	–	336 ± 10.1 ^1^	–
II	2.87	439.0 ± 4.0	450.9 ±4.5	306.5 ± 9.2	329.6 ± 10.0
III	8.13	198.7 ± 2.2	222.6 ± 2.3	154.1 ± 5.4	175.2 ± 5.3
IV	4.06	386.4 ± 3.8	429.1 ± 4.1	281.5 ± 8.2	323.5 ± 9.4

^1^ The biogas and methane yield was calculated for a period of 14 days.

**Table 7 ijerph-20-00969-t007:** Change in the elemental composition of the substrate at varying OLRs.

Stages	OLR, Kg VS/m^3^ d	C_before_, g/Kg TS	N_before_, g/Kg TS	H_before_, g/Kg TS	S_before_, g/Kg TS	ΔC ^1^	ΔN ^1^	ΔH ^1^	Δ S ^1^
I	–	650.0 ±19.5	25.0 ±0.1	43.7 ±0.2	15.3 ±0.1	10.0 ±0.03	0.3 ±0.001	0.5 ±0.002	0.2 ±0.001
II	2.87					15.1 ±0.03	0.4 ±0.003	0.7 ±0.003	0.3 ±0.001
III	8.13					12.5 ±0.04	0.2 ±0.001	0.5 ±0.007	0.2 ±0.001
IV	4.06					13.0 ±0.04	0.3 ±0.002	0.6 ±0.002	0.3 ±0.001

^1^ Results are presented in units: g/Kg TS d.

## Data Availability

Not applicable.

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
