# Peer review of "Research on Biogas Yield from Macroalgae with Inoculants at Different Organic Loading Rates in a Three-Stage Bioreactor"

_ijerph, 2023, doi:10.3390/ijerph20020969_

Round 1

Reviewer 1 Report

Alvydas Zagorskis et al. in their manuscript presents study for improvement of TSB operating parameters and determination of the methane yield of digesters using different organic loading rates (OLR).  Topic of manuscript are highly implemented in research area; therefore, this work could have valuable input in such research area. However, I have some comments/questions, which should be revised before publication:

1.      The novelty of work is not highlighted.  

2.      In my opinion introduction part is too long. I recommend to add reference number after citing certain authors (lines 128-130; 131-132);

3.      “A number of studies have been conducted using marine macroalgae. Bioreactors of periodic operation of small capacity, the volume of which can reach from 500 mL to 2500 mL, are usually used for conducting research [16;29;30]. The aim of the research is to conduct biogas yield and qualitative composition studies at different OLR when macroalgae with additives are used in a three-stage bioreactor. The aim of the research is to assess how biogas yield, quality, and substrate OM depend on OLR. During the research, it was determined how the biogas yield changed and how long it took to stabilize when changing the OLR”. English and construction of sentences must be checked.

Aim is not clear highlighted. What is the aim of this work: “The aim of the research is to conduct biogas yield and qualitative composition studies at different OLR when macroalgae with additives are used in a three-stage bioreactor” or  “The aim of the research is to assess how biogas yield, quality, and substrate OM depend on OLR”?

4.      146 line. From which river? Should be written exactly.

5.      156 line. Is mentioned that elemental composition of each biomass was determined. With which equipment?

6.      227-228 line which equipment was used for determination of composition?

7.      I suggest firstly add analytical methods parts and then part of information about composition.

8.      “For biogas composition GasData biogas analyzer GFM410 was used”. The errors of device look quite huge and such equipment is more suitable for industrial purposes. Therefore, Gas Chromatography would be much more reliable and appropriate.

9.      In 267-267 lines is presented the aim. In abstract, in introduction part and in mentioned lines aim of study looks different. Aim should be highlighted.

10.  In 279-280 lines sentences must be combined.

11.  281 line: “The increase was due to the methanogenic bacteria receiving nutrients”. Can you confirm it with experimental data?

12.  301 line: “Methanogenic bacteria do not manage to completely break down the substrate VS”. More deeply explanation should be added, answering to the question why?  

13.  351-352 lines: “Studies found that oxygen (O2) and hydrogen sulfide (H2S) concentrations were low and at all stages of the experiment reached 0-2% and 0-10 ppm respectively”. How you can exactly determine this values if errors of equipment are 0.5% for O2 and 5% for H2S in the range 0-1500ppm (errors 0-75ppm). Therefore, reliability of obtained values should be considered. Gas chromatography here would be much more appropriate.

14.  In results part obtained results are presented, however more deeply explanation why changing OLR influence yields of methane is missed.

15.  In discussion part more deeply discussion about obtained results also could be performed. In discussion part literature data is added, however discussion about obtained data, processes which occurs in the reactor is not explained. Maybe some changes in concentration of bacteria, or pH can occur, which can also influence the performance of bioreactor?  I suggest to discuss more about obtained results and compare them with other scientific data.

16.  In 462-476 lines part about C:N ratio is added, however in this this study, C:N ratio was not discussed, therefore this part is not significant in this part. It can be mentioned in the literature part.

17.   In Discussion part and literature part some information is repeated, therefore should be carefully revised.

 After careful revision of my comments and revision of work, manuscript can be accepted for publication.

Author Response

Thank you for your relevant comments, we are sending comments. The corrections are shown in blue in the text.

  1. Taking into account the remark, the novelty of the work was specified and emphasized more in the annotation, introduction and conclusions of the work. (lines 18-21; 124-128; 566-568).
  2. Taking into account the remark, the introduction was shortened by removing part of the text, leaving only the text highlighting the essential relevance of the work.
  3. Taking into account the remark, the purpose of the work was clarified. Some sentences related to the purpose of the work were omitted (lines 124-128).
  4. Taking into account the remark, the research methodology was supplemented by indicating the name of the river (line 134).
  5. Taking into account the remark, the research methodology was supplemented by indicating the equipment (lines 151-152).
  6. Taking into account the remark, the research methodology was supplemented by indicating the equipment (line 231).
  7. Based on the remark, analysis methods were added to the set parameters (lines.142-143; 151-152; 226-228; 231). Also, the analysis methods were detailed in subsection 2.4.
  8. We agree that gas chromatography would be a more accurate research method. During the biogas composition studies, attention was paid to the accuracy of the results. In order to increase the reliability of the results, the device was calibrated before the measurement. Data were checked periodically using a gas chromatograph. The manufacturer of the device is known, one of the leaders of such analyzers in the world, so we believe that the data is reliable.
  9. Taking into account the remark, the purpose of the work was unified in the abstract and introductory part (lines 18-21; 125-128). In the results section, the sentence about the purpose of the work was dropped.
  10. According to the remark, the sentences were joined (lines 287-288).
  11. In response to the comment, line 289-290 has been revised to indicate an assumption that may have led to the increase.
  12. Based on the remark, the line was supplemented with information indicating what could have led to such a breakdown (lines 313-315).
  13. We agree that gas chromatography would be a more accurate research method. During the biogas composition studies, attention was paid to the accuracy of the results. In order to increase the reliability of the results, the device was calibrated before the measurement. Data were checked periodically using a gas chromatograph. The manufacturer of the device is known, one of the leaders of such analyzers in the world, so we believe that the data is reliable.
  14. Taking into account the comment, the results section was supplemented with information about the influence of OLR on methane yield (lines 391-401).
  15. Based on the comment, the discussion section was supplemented.
  16. In response to the comment, some information about C:N has been moved from the discussion section to the introduction section (lines 91-93). Some information about C:N has been dropped in the discussion section.
  17. The discussion and introduction sections have been revised in light of the comment. Duplicate information has been removed from the introduction. General information about C:N has been moved to the introduction section (lines 91-93).

Reviewer 2 Report

This manuscript presents an technical approach to modern research on biofuels.

The research was carried out on a common species that is not under protection and with low culturing requirements, which may have an application value in the future.

The methods seem to be sound. I found the manuscript quite well-written and interesting however some points need to be amended.  

Specific comments: 

1.      few-sentenced paragraphs for example 93-95, 121-123; to be connected to the rest of the text, 351-352, 454-456

-they sound like sentence equivalents, leaving the reader without explanation

2.      the introduction to the subject is clear, but it lacks information why the authors chose this species and maybe more information about it, is it used in this type of research for the first time; is it something new in this part of Europe, etc.?

3.      The manuscript requires minor editorial corrections, as in few places there are no spaces where they should be, or they are double.

4.      Line 234: instead of ‘the’ change to ‘this’

5.      Line 236: Device error ±0.002; equals/is ±0.002?

6.      Some paragraphs needs to be re-written by native to be more clear, for example 267 -269

7.      remove the frame surrounding the graph-figure 5, figure 6

8.      the style of lines in table 6, table 7 should be unified

9.      the style of bars in figure 6 should be changed to be more scientific

10.  the discussion section discusses the results of other work but without context to the work done within this manuscript

11.  line 455: italics missing M. pyrifera

12.  lines 460-461:The Zhong bioreactor used blue macroalgae.
Using these macroalgae was about 50%

- and the results of this is….?

And many, many other sentences do not lead to any development

13.  I don’t think the paragraphs should begin like this:

Pomdaeng et al. sought to utilize …..

Researcher Sun and others…..

14.  there is little discussion especially in the context of comparisons with this study

The cited literature is quite new, which only emphasizes the importance of the topic. Any results from this topic are important and the authors should do their best to improve this manuscript.

In spite of my generally positive opinions, the manuscript needs the magic wand of a language specialist because despite generally correct English, it is hard to read.

Despite some flaws, I recommend accepting this manuscript after minor revision, which improves the quality of the paper. 

Author Response

Thank you for your relevant comments, we are sending comments. The corrections are shown in blue in the text.

1. In light of the comment, the text has been revised and paragraphs of several lines have been added to the rest of the text. Some lines have been removed;

2. Taking into account the comment, the article was supplemented with information about the choice of species (lines 134-138);

3. The article has been reviewed for editorial corrections;

4. Taking into account the remark, the line was clarified (line 235);

5. Taking into account the remark, the line was clarified (line 238);

6. Taking into account the remark, the text was corrected, and the text of little significance (including lines 267-269) was removed;

7. Considering the remark the frame surrounding the graph-figure 5, figure 6 was removed;

8. Line style was unified in table 6 and table 7;

9. In view of the remark Figure 6 has been improved;

10. Based on the comment, the discussion section has been improved;

11. Taking into account the remark, the line was clarified (lines 500-501);

12. Based on the comment, the discussion section was improved, the paragraphs were corrected;

13. Based on the comment, the discussion section was improved, the paragraphs were corrected;

14. Based on the comment, the discussion section has been improved.

Reviewer 3 Report

The Manuscript ID: ijerph-2118066 Research on biogas yield from macroalgae with inoculants at different organic loading rates in a three-stage bioreactor” requires revision before accepted for publication. The specific comments are given below.

1.     Provide significant words which are more relevant to the work in a logical sequence as ‘keywords’.

2.     The "Introduction" section should follow the state of the art of this field and review what has been done, for supporting the research gap and the significance of this study. Please improve the state of the art overview, to clearly show the progress beyond the state of the art. Give examples of fermentation of microalgae with other substrates, e.g.

https://doi.org/10.3390/en13092186

https://doi.org/10.1016/j.bej.2016.11.008

https://doi.org/10.3389/fenrg.2019.00111

https://doi.org/10.1007/s13399-022-03587-7

https://doi.org/10.1016/j.enconman.2018.04.032

https://doi.org/10.1016/j.ijhydene.2018.06.069

https://doi.org/10.3390/app12147291

https://doi.org/10.1016/j.rser.2019.05.061

https://doi.org/10.1016/j.chemosphere.2020.128963

3.     In the last paragraph of the introduction, clearly indicate the research hypothesis.

4.     Complete the units, e.g. Ln 24, 25.

5.     Take care of the nomenclature, e.g. Methanobacterium sp. write "sp." no italics.

6.     Why was Cladophora glomerata used in the research? What was the criterion for choosing this green macroalgae?

7.     Ln 156 “the elemental composition of each biomass was determined” – specify the analytical methodology, apparatus, etc.

8.     Discuss exactly how such OLR values were determined.

9.     Please indicate the manufacturer, city, country when mentioning the equipment.

10.  Statistical research is very important in experiments. How were the significances of the differences between the variables determined?

11.  I am asking you to perform an energetic analysis in order to assess the legitimacy.

12.  It is worth thinking about planning DOE experiments and presenting surface effects on biogas/methane production.

13.  More details should be included in the conclusion.

14.  It is also recommended to discuss and explain what should be the appropriate policies based on the findings of this study.

15.  Update your literature review with the latest publications from 2019-2022.

Author Response

Thank you for your relevant comments, we are sending comments. The corrections are shown in blue in the text.

1. Taking into account the remark, the significant words have been specified and those that are more related to the work are presented in a logical sequence;

2. Taking into account the remark, the introduction section was supplemented with recommended literature (lines 43; 48-51; 58-61; 68-75; 76-86; 105-108; 120-123);

3. Considering the remark, the introduction section was supplemented with hypotheses (lines 124-125);

4. Taking into account the remark, the units were specified (lines 24-25);

5. Taking into account the remark, the nomenclature was revised and corrected;

6. Taking into account the remark, the research methodology was supplemented by indicating the selection criteria for Cladophora glomerata (lines 134-138);

7. Taking into account the remark, information was provided about the elemental composition determination equipment (lines 151-152);

Taking into account the comment, the methodological part of the article has been supplemented with information about the OLR determination (lines 205-209; 244-247). An additional equation is presented (Equation No. 2);

9. Considering the remark in subsection 2.4, information about the manufacturer, city, country when mentioning the equipment (lines 235-239; 260-263) was provided for the devices;

10. Considering the remark information was provided about the significances of the differences between the variables determined in subsection 2.4. (lines 271-273);

11. Taking into account the remark, the results part was supplemented with energy analysis (lines 447-458);

12. Table 5 on experiment planning (steps) was clarified and additional information was provided on the energy costs of the three-stage bioreactor and the amount of energy produced (lines 447-458);

13. Taking into account the remark, the conclusions were supplemented with information about the efficient use of biomass and the sustainable policy of using natural resources to extract energy (lines 558-564; 583-588);

14. Taking into account the comment, the discussion section was supplemented with information about the efficient use of biomass and the sustainable policy of using natural resources for energy extraction (lines 548-556);

15. Considering the remark, the literature was supplemented/updated with recommended and additional recent literature publications.

Round 2

Reviewer 3 Report

Thank you.